# Palygorskite-Based Organic–Inorganic Hybrid Nanocomposite for Enhanced Antibacterial Activities

**DOI:** 10.3390/nano11123230

**Published:** 2021-11-28

**Authors:** Aiping Hui, Fangfang Yang, Rui Yan, Yuru Kang, Aiqin Wang

**Affiliations:** Key Laboratory of Clay Mineral Applied Research of Gansu Province, Center of Eco-Material and Green Chemistry, Lanzhou Institute of Chemical Physics, Chinese Academy of Sciences, Lanzhou 730000, China; aphui1215@163.com (A.H.); yangff@licp.cas.cn (F.Y.); yanjie4712010@163.com (R.Y.); yurukang@licp.cas.cn (Y.K.)

**Keywords:** plant essential oils, ZnO, palygorskite, carvacrol, antibacterial activities

## Abstract

A synergistic antibacterial strategy is effective in enhancing the antibacterial efficacy of a single antibacterial material. Plant essential oils (PEOs) are safe antibacterial agents. However, some of their characteristics such as intense aroma, volatility, and poor thermal stability limit their antibacterial activity and applications. In this paper, five kinds of PEOs were incorporated onto ZnO/palygorskite (ZnO/PAL) nanoparticles by a simple adsorption process to form organic–inorganic nanocomposites (PEOs/ZnO/PAL) with excellent antibacterial properties. TEM and SEM analyses demonstrated that ZnO nanoparticles uniformly anchored onto the surface of rod-like PAL, and that the structure of ZnO/PAL maintained after the incorporation of ZnO nanoparticles and PEOs. It was found that carvacrol/ZnO/palygorskite (CAR/ZnO/PAL) exhibited higher antibacterial activities than other PEOs/ZnO/PAL nanocomposites, with minimum inhibitory concentration (MIC) values of 0.5 mg/mL and 1.5 mg/mL against *Escherichia coli* (*E. coli*) and *Staphylococcus aureus* (*S. aureus*), respectively. Moreover, the antibacterial efficiency of CAR/ZnO/PAL nanocomposites was superior to that of ZnO/PAL and pure CAR, demonstrating the synergistic effect that occurs in the combined system. PAL serving as a carrier for the combination of organic PEOs and ZnO nanoparticles is an effective strategy for enhanced, clay-based, organic–inorganic hybrid antibacterial nanocomposites.

## 1. Introduction

Plant essential oils (PEOs) and their extracts have been examined for their effectiveness in food preservation [1,2,3,4,5]. PEOs as naturally occurring, biologically active agents have been shown to possess anti-inflammatory, antioxidant, antibacterial, and antifungal properties [6,7,8]. Recently, there has been a growing demand for natural PEOs in antibacterial applications owing to their safety and extensive sources such as plant leaves, barks, stems, roots, flowers, and fruits [9,10,11,12,13]. The good antibacterial activity of PEOs may be attributed to the hydrophobicity that could damage the bacterial cell membrane and cause the leakage of the internal contents of the cells [3,14,15,16]. Nevertheless, PEOs also suffer from the problems of volatility, tangy aromatic odor, and poor thermal stability, which would diminish their antibacterial activity as well as limit their practical applications [17,18]. A lot of research has demonstrated that PEO-based nanocomposites have been designed to overcome these issues. In addition, PEOs show more sensitivity toward Gram-positive bacteria than Gram-negative bacteria. Building microencapsulation or emulsion is a classical method of combining a variety of PEOs, while another effective strategy is to composite inorganic metal oxide with PEOs [19,20]. Furthermore, it is known that the complexes of various antibacterial components could obtain synergistic antibacterial effects, thus making it a feasible route for utilizing the advantages of various antibacterial factors. 

Metal oxide such as ZnO, MgO, and TiO_2_ are all very efficient in inhibiting the growth of bacteria, and these antibacterial materials show better antibacterial properties against Gram-negative bacteria than Gram-positive bacteria [21,22,23,24,25]. Therefore, taking advantage of PEOs and metal oxide may be a facile approach to the building of synergistic antibacterial composites [3,6,26]. Among these metal oxides, ZnO exhibits excellent antibacterial activity toward foodborne pathogens such as *Escherichia coli* (*E. coli*) and *Staphylococcus aureus* (*S. aureus*) [21,22,27,28,29].

Interestingly, ZnO nanoparticles (NPs) can be loaded onto clay-based carriers, which could effectively avoid the agglomeration of ZnO NPs [30,31,32,33]. Palygorskite (PAL), as a naturally available one-dimensional nanomaterial, a special crystal structure, stacking mode, and nanometric dimension, endowing it with plentiful pores, a high aspect ratio, good ion-exchange capacity, and affluent surface groups [34,35,36]. PAL as a carrier could load inorganic NPs as well as immobilize organic molecules. Yang’s group and Osajima’s group have reported ZnO/PAL nanocomposites and found that the antibacterial properties of ZnO against *E. coli* could be improved through conducting nanocomposites [32,37]. In our previous work, ZnO/PAL were prepared in the presence of surfactants using an easy-to-operate hydrothermal method and chemical deposition [31,38]. Incorporating PEOs onto ZnO/PAL nanoparticles is expected to obtain highly efficient and broad-spectrum antibacterial materials.

In this paper, an ultrasonic-assisted dipping method was used to incorporate PEOs onto a ZnO-loaded PAL nanoparticles to build organic–inorganic nanocomposites with broad-spectrum antibacterial capabilities. PAL as a carrier can effectively immobilize PEOs and ZnO NPs, reducing the volatility of PEOs and showing synergistic antibacterial performance. The preparation parameters of ZnO/PAL such as the variety of alkaline, amount of PAL, reaction temperature, and time were optimized based on antibacterial properties. The antibacterial activities of PEOs/ZnO/PAL nanocomposites formed by different PEOs were evaluated, and the synergistic antibacterial effects and synthesis mechanism were also discussed.

## 2. Materials and Methods

### 2.1. Materials

PAL originated from the Huangnishan Mine and was provided by Huida Mineral Technology Co. Ltd., Huaian, China. It was crushed and purified by 2% H_2_SO_4_ solution at the solid/liquid ratio of 1:10 to remove the associated carbonates. The purified PAL was filtered by passing it through a 200-mesh sieve for further use. The chemical composition is as follows: SiO_2_, 57.39%; Al_2_O_3_, 8.41%; MgO, 12.21%; Fe_2_O_3_, 5.05%; Na_2_O, 2.03%; K_2_O, 0.9%; CaO, 1.35%, and the value of specific surface area was 209.3 m^2^/g. Zinc nitrate hexahydrate (Zn(NO_3_)_2_·6H_2_O, 99.0%), sodium hydroxide (NaOH, 96.0%), and ethanol were purchased from Tianjin Kermel Chemical Regent Co., Ltd., Tianjin, China). The PEOs of citral (97%, cis + trans), thymol (>99%), carvacrol (CAR, 99%), oregano oil (carvacrol 85%, thymol 6%), and cinnamaldehyde (98%) were bought from Shanghai Macklin Biochemical Co., Ltd. (Shanghai, China). The chemicals were directly used without further purification, and distilled water was used throughout the experiment.

### 2.2. Preparation of PEOs/ZnO/PAL

The PEOs were incorporated onto PAL and ZnO/PAL by a simple dipping method. In order to synthesize the PEOs/PAL nanocomposites, 2 g of purified PAL was added to a PEO ethanol solution (10%, *w*/*v*), and the mixture was ultrasonically dispersed for 30 min and shocked for 24 h with 160 r/min. The powder was subsequently collected by centrifugation and dried at 45 °C for 12 h in an oven. CAR/PAL nanocomposites with various concentrations of CAR (1%, 2.5%, 5%, 10%, 20%) were synthesized by the same process. ZnO/PAL nanoparticles were prepared by chemical deposition and calcination method (Hui et al., 2020; Huo et al., 2010); the preparation conditions and antibacterial activities of ZnO/PAL nanoparticles are shown in Appendix A.

### 2.3. Characterization

The chemical composition of PAL was determined using a Minipal 4 X-ray fluorescence spectrometer (PANalytical, The Netherlands, Germany). Morphological evolution of the samples was observed using a field emission scanning electron microscopy (SEM, JSM-6701F, JEOL, Tokyo, Japan) and a transmission electron microscope (TEM, JEM-1200EX, JEOL, Tokyo, Japan). X-ray diffraction (XRD) patterns were acquired on an X’ Pert PRO diffractometer equipped with a Cu Kα radiation source, 40 mA, and 40 kV at a scanning rate of 0.02° per second. Fourier Transform infrared (FTIR) spectra were measured on a Nicolet NEXUS spectrometer (Thermo Fisher Scientific, Waltham, Massachusetts, America) in the range of 4000–400 cm^−1^ using potassium bromide pellets. Nitrogen adsorption–desorption isotherms were recorded using the Brunauer–Emmett–Teller analysis (BET, Micromeritics, Norcross, Georgia, America). Thermogravimetric analyses (TG) were performed on the simultaneous thermal analyzer (STA6000, PerkinElmer, America), operating from 30 °C to 800 °C, with a heating rate of 10 °C/min in a nitrogen atmosphere.

### 2.4. Antibacterial Assay

Antibacterial activities of the samples were evaluated by examining the minimum inhibitory concentration (MIC) value against Gram-negative *E. coli* (CVCC1524, C83698, O8: K91) and Gram-positive *S. aureus* (CVCC1885, C56005) bacteria [31], which were kindly provided by China Veterinary Culture Collection Center. Bacterial strains were isolated from the Luria–Bertani agar plate, and added into fresh Luria–Bertani broth separately for incubation at 37 °C in a shaking incubator, at a speed of 160 r/min for 12 h. The strains were inoculated into the fresh medium again and cultivated for 3 h in the logarithmic phase, at a volume ratio of 1:100 of bacteria liquid and fresh medium. Then, a certain amount of sample was mixed with 40 mL of the Macconkey medium, and 1 μL fresh bacteria liquid (10^8^ CFU/mL) was put into the medium with three parallel dots at different locations, repeated on three parallel plates for each sample. The tested plates were kept in an incubator at 37 °C for 24 h. Positive control with the exclusion of antibacterial materials on an agar plate and a blank control with no samples and bacteria liquid dots were performed under the same cultivation conditions. The MIC value was reported as the lowest concentration to completely inhibit the growth of each bacterial strain being tested.

## 3. Results and Discussion

### 3.1. Possible Formation Mechanism of PEOs/ZnO/PAL

PEOs such as citral, thymol, CAR, oregano oil, and cinnamaldehyde were selected as natural antibacterial factors. These PEOs are all small molecules with active groups such as the phenolic hydroxyly group, carbonyl group, and aldehyde group (Figure 1a), and favor adsorption onto the surface of PAL and ZnO/PAL through hydrogen bonding and weak electrostatic interaction, or by entering into the nanopores of PAL. Two steps for loading ZnO NPs and PEOs onto PAL are shown in Figure 1b. The electrostatic interaction between PAL with a negative charge and Zn^2+^ with a positive charge could make them self-assemble together in an aqueous solution. When Zn^2+^ was introduced into the reaction mixture, it assembled on the PAL surface spontaneously and uniformly. By adjusting the pH of the solution, the sheet-like Zn(OH)_2_ appeared and finally transformed into ZnO/PAL after calcination. For- PEOs/ZnO/PAL nanocomposites, PEO molecules were incorporated onto ZnO/PAL via a simple dipping method. PEOs with the phenolic hydroxyly group, carbonyl group, and aldehyde group could be connected with Si–OH groups on the outer surface of ZnO/PAL by hydrogen-bond interaction. In these PEOs/ZnO/PAL composites, PEOs play a vital role in enhancing the antibacterial activities of inorganic ZnO/PAL nanoparticles. Small CAR molecules displayed better antibacterial activities toward two model bacteria than other PEOs; therefore, the following work was conducted to obtain the reasonable concentrations of CAR, as shown in Figure 1b.

### 3.2. FTIR

In the spectrum of PAL shown in Figure 2a, the bands at 3550 cm^−1^, 3430 cm^−1^, and 1636 cm^−1^ are related to the stretching vibration of the coordinated water, the stretching vibration, and the antisymmetric stretching vibration of zeolitic water and adsorption water [39], while the characteristic absorption bands at 1020 cm^−1^ and 476 cm^−1^ were attributed to the stretching vibration of Si–O–Si and δ_Si–O_, respectively [35,40,41]. In the case of PEOs/ZnO/PAL, the band at 3740 cm^–1^ disappeared and shifted to lower wavelengths, and a broad band appeared from 3700 cm^–1^ to 3400 cm^–1^ (Figure 2a), which indicated that the interactions between ZnO/PAL and PEOs had occurred, and that the new absorption peaks that appeared at 1660 cm^−1^ corresponded to the C=C or benzene ring stretching vibrations of the PEOs. The FTIR spectra of PAL containing different amounts of CAR are shown in Appendix A and Figure 2b, the antisymmetric stretching vibration absorption bands of the CH_3_ group and the CH_2_ group were located at 2968 cm^–1^ and 2856 cm^–1^, respectively, and the symmetric stretching vibration absorption band of the CH_3_ group was found at 2870 cm^–1^ [42]. Moreover, the absorption bands at 1400–1600 cm^−1^, 1186 cm^−1^, and 800–900 cm^−1^ can be ascribed to the aromatic C=C stretching vibration, aromatic O–H stretching vibrations peak, and aromatic C–H bending, respectively [43].

### 3.3. XRD Patterns

XRD patterns of PAL, ZnO/PAL, and CAR/ZnO/PAL are shown in Figure 3. The diffraction peaks at 2*θ* of 8.4°, 13.7°, 16.3°, 19.8°, 19.9°, and 34.4° correspond to (110), (200), (130), (040), (310), and (102) planes of PAL, respectively [31,35,40]. It is worth noting that 2*θ* = 26.7° shows a characteristic peak for quartz [40], and the diffraction peaks of ZnO/PAL were located at 2*θ* = 31.7°, 34.4°, 36.3°, 47.6°, 56.6°, 62.9°, 66.5°, 67.9°, and 69.1°, which corresponded to (100), (002), (101), (102), (110), (103), (200), (112), and (201) of a wurtzite ZnO [21]. After the incorporation of CAR, the CAR/ZnO/PAL nanocomposites displayed a similar XRD pattern to that of ZnO/PAL, and there was an obvious change in the intensity of the diffraction peaks for PAL, which may be due to the weak interface interaction of the PEOs with ZnO/PAL.

### 3.4. TG Curves

The weight loss of PAL, ZnO/PAL, and CAR/ZnO/PAL was also demonstrated by the TG curves, as shown in Figure 4, where the weight loss of PAL could be divided into four steps [44]. The first step is below 100 °C, owing to the loss of superficially adsorbed water and most of the zeolitic H_2_O. In the second step, the residual zeolitic H_2_O and the first half of the structural water molecules were removed from 100 °C to 250 °C. In the third step, the loss of the residual first half of the structural water and the second half of the structural water were removed from 500 °C to 550 °C, and the removal of the second half of the structural water and the hydroxyl groups was observed in the last region at about 550–800 °C. In terms of ZnO/PAL, the weight loss is lower in comparison with PAL; in addition, it can be seen that the temperature increased to 300 °C, the CAR molecules began decomposition with the growing temperature. There are four stages of weight loss for the CAR/ZnO/PAL nanocomposites, which corresponding to the weight loss of PAL, as shown in the insert image of Figure 4. CAR molecules escaped from the surface of PAL when the temperature was at 105 °C. The maximum decomposition temperature was 315 °C, which could have been caused by the small CAR molecules that loaded onto PAL. The total weight loss of CAR/ZnO/PAL was 8.2%.

### 3.5. SEM and TEM

Figure 5 shows the SEM images of PAL, ZnO/PAL, and CAR/ZnO/PAL; it can be seen that PAL exhibits a rod-like morphology (Figure 5a), and ZnO NPs anchored uniformly onto the surface of PAL (Figure 5b). Moreover, after the incorporation of the CAR molecules, the morphology of ZnO/PAL nanocomposites had no clear change, but the nanorods had become loose due to the active CAR molecules adsorbed on the surface of ZnO/PAL through the hydrogen bond interaction (Figure 5c); the interfacial repulsion between the CAR molecules caused the CAR/ZnO/PAL nanocomposites to become looser. TEM was carried out to further investigate the changes in the microstructures of CAR/ZnO/PAL. As shown in Figure 6, the length of the rod-like PAL typically varies from 0.5 μm to 1.5 μm, with a diameter from 20 nm to 70 nm (Figure 6a). Meanwhile, it can be clearly observed that a number of ZnO NPs with a diameter of about 40 nm were incorporated onto the surface of rod-like PAL (Figure 6b). The morphology of ZnO/PAL was maintained after combining with the CAR molecules, confirming that ZnO NPs were loaded tightly onto the surface of PAL (Figure 6c). Moreover, compared to the previous work of mechanical milling, the dipping method is more favorable for the loading of small CAR molecules onto PAL or ZnO/PAL, and the rod-like structures are kept very well during the process; in contrast, the structure is easy to damage using the mechanical milling method [45].

### 3.6. BET Analysis

As shown in Table 1, the microporous surface area (*S*_micro_) of the CAR/ZnO/PAL nanocomposites were disappeared when the CAR molecules incorporated on PAL. The *S*_micro_, *S*_ext_, and *V*_total_ of the CAR/ZnO/PAL nanocomposites extremely decreased with the increasing concentrations of CAR. The specific surface area of PAL was 125 m^2^/g, but it decreased to 27 m^2^/g for PAL/ZnO, and decreased from 25 m^2^/g to 9 m^2^/g for the CAR/ZnO/PAL nanocomposites with growing concentrations of CAR. The extra surface area (*S*_ext_) also decreased with the addition of CAR. It may be due to the existence of ZnO NPs or small CAR molecules on the surface or channel of PAL, which set a big barrier for N_2_ molecules entering into the nanopore structure [45]. With the increasing amount of CAR, *S*_BET_ and *S*_ext_ obviously decreased, showing that CAR molecules could be adsorbed onto the surface of ZnO/PAL or enter into the nanopores of PAL due to the hydrogen bond interaction between the phenolic hydroxyl group of CAR and the silanol group of PAL. 

### 3.7. Antibacterial Evaluation

*E. coli* and *S. aureus* were chosen to evaluate the antibacterial activities of the samples, including PAL, PEOs/PAL, ZnO/PAL, and PEOs/ZnO/PAL. The MIC value was defined as the lowest concentration (mg/mL) of the tested sample preventing the visible growth of a microorganism under defined conditions. The MIC results of PAL are shown in Appendix A, which were more than 50 mg/mL toward *E. coli* and *S. aureus*. As a control, PAL did not show obvious antibacterial activities against Gram-negative *E. coli* and Gram-positive *S. aureus*, and there were also no visible differences in the MIC results with different amounts of PAL toward the two model bacteria. It can be clearly observed that the microbial colony of the bacteria disappeared with the treatment of ZnO/PAL (Figure 7), which indicated that ZnO/PAL exhibited good antibacterial activities against *E. coli* and *S. aureus*; the MIC values toward *E. coli* and *S. aureus* were 1.5 mg/mL and 2.5 mg/mL, respectively. 

Five kinds of PEOs were selected to be combined with ZnO/PAL in order to enhance the broad-spectrum antibacterial activities, as shown in Figure 8 (*E. coli*) and Figure 9 (*S. aureus*). The bacteria colonies of *E. coli* and *S. aureus* disappeared clearly upon contact with the PEOs/ZnO/PAL nanocomposites. In particular, five kinds of PEOs combined with ZnO/PAL nanoparticles had different antibacterial properties against the ´Gram-negative and Gram-positive bacteria. The CAR/ZnO/PAL nanocomposites and oregano oil/ZnO/PAL nanocomposites displayed better antibacterial properties toward *E. coli* than other PEOs/ZnO/PAL nanocomposites (Figure 8), while the CAR/ZnO/PAL nanocomposites also exhibited higher antibacterial efficacy against *S. aureus* (Figure 9) from among the PEOs/ZnO/PAL nanocomposites. The differences can be attributed to the distinct composition of the cell membrane and the cell wall as well as the functional groups present in different kinds of PEOs and the synergistic effects induced by the interactions of PEOs and ZnO/PAL nanoparticles [6,38,46]. The MIC value of the CAR/ZnO/PAL nanocomposites against *E. coli* and *S. aureus* reached 0.5 mg/mL and 1.5 mg/mL, respectively (Appendix A).

Meanwhile, the antibacterial activities of the orange oil/ZnO/PAL nanocomposites against *E. coli* were the same as those of the CAR/ZnO/PAL nanocomposites due to the fact that CAR was the main component of orange oil. The antibacterial action of CAR on bacteria occurs via the change in cell membrane permeability, which results in the depletion of the intracellular ATP pool, ultimately leading to cell death [47,48]. In order to determine the superior concentration of CAR among the CAR/ZnO/PAL nanocomposites, the MIC value of the CAR/ZnO/PAL nanocomposites prepared with different concentrations of CAR against *E. coli* is shown in Figure 10, and the MIC evaluation of PAL with different concentrations of CAR is shown in Appendix A. It can be seen that when the concentration of CAR was maintained at 10%, the CAR/ZnO/PAL nanocomposites reached their best antibacterial activity. When the concentration of CAR further increased to 20%, the antibacterial activities did not increase, which was possibly caused by the effective amounts loaded onto ZnO/PAL. The antibacterial performance of the CAR/ZnO/PAL nanocomposites was attributed to the synergistic antibacterial effect, which can be described as follows. The nanoparticles and reactive oxygen species generated by ZnO such as superoxide, singlet oxygen, and hydroxyl radicals could damage the structural integrity of bacteria [49,50]. Meanwhile, the slow release of CAR molecules from the nanocomposites can disturb the formation of nucleic acids, protein, or cell wall synthesis [47,49,51]. Therefore, the effects of physical damage combined with the chemical adsorption and bactericidal function would create broad-spectrum antibacterial effects. In particular, when compared with the previous antibacterial materials prepared with a PAL carrier, the antibacterial activities of the CAR/ZnO/PAL nanocomposites were superior to those of single ZnO/PAL nanoparticles (1.5 mg/mL) and CAR/PAL hybrid materials (2 mg/mL) toward Gram-negative *E. coli* [31,43].

## 4. Conclusions

A series of organic–inorganic antibacterial nanocomposites with broad-spectrum antibacterial effects were prepared based on natural PEOs and ZnO/PAL nanoparticles using a dipping method with ultrasonic processing. ZnO nanoparticles with a diameter of 40 nm uniformly anchored onto the surface of rod-like PAL. TG analysis indicated the effective loading content of CAR at 8.2%. The antibacterial activities did not increase when the concentration of CAR further increased to 20%. The obtained PEOs/ZnO/PAL nanocomposites exhibited synergistic antibacterial activities against *E. coli* and *S. aureus*. Among these nanocomposites, the CAR/ZnO/PAL nanocomposites exhibited better antibacterial efficacy than other PEOs/ZnO/PAL nanocomposites, with MIC values of 0.5 mg/mL and 1.5 mg/mL toward *E. coli* and *S. aureus*, respectively. This study provides a feasible route for the preparation of broad-spectrum antibacterial materials with PEOs and inorganic antibacterial carriers.

## Figures and Tables

**Figure 1 nanomaterials-11-03230-f001:**
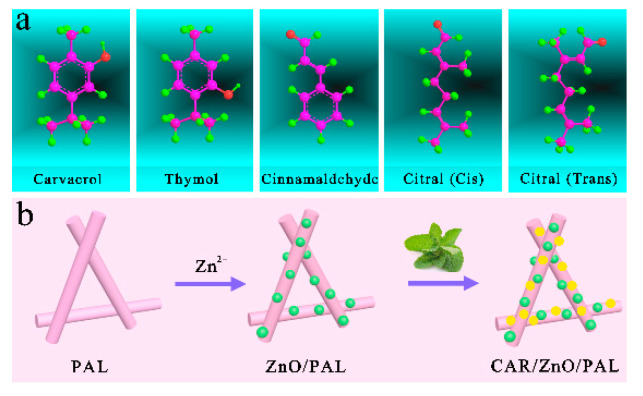
(**a**) Chemical structures of PEOs, (**b**) schematic illustration for the preparation of CAR/ZnO/PAL nanocomposites.

**Figure 2 nanomaterials-11-03230-f002:**
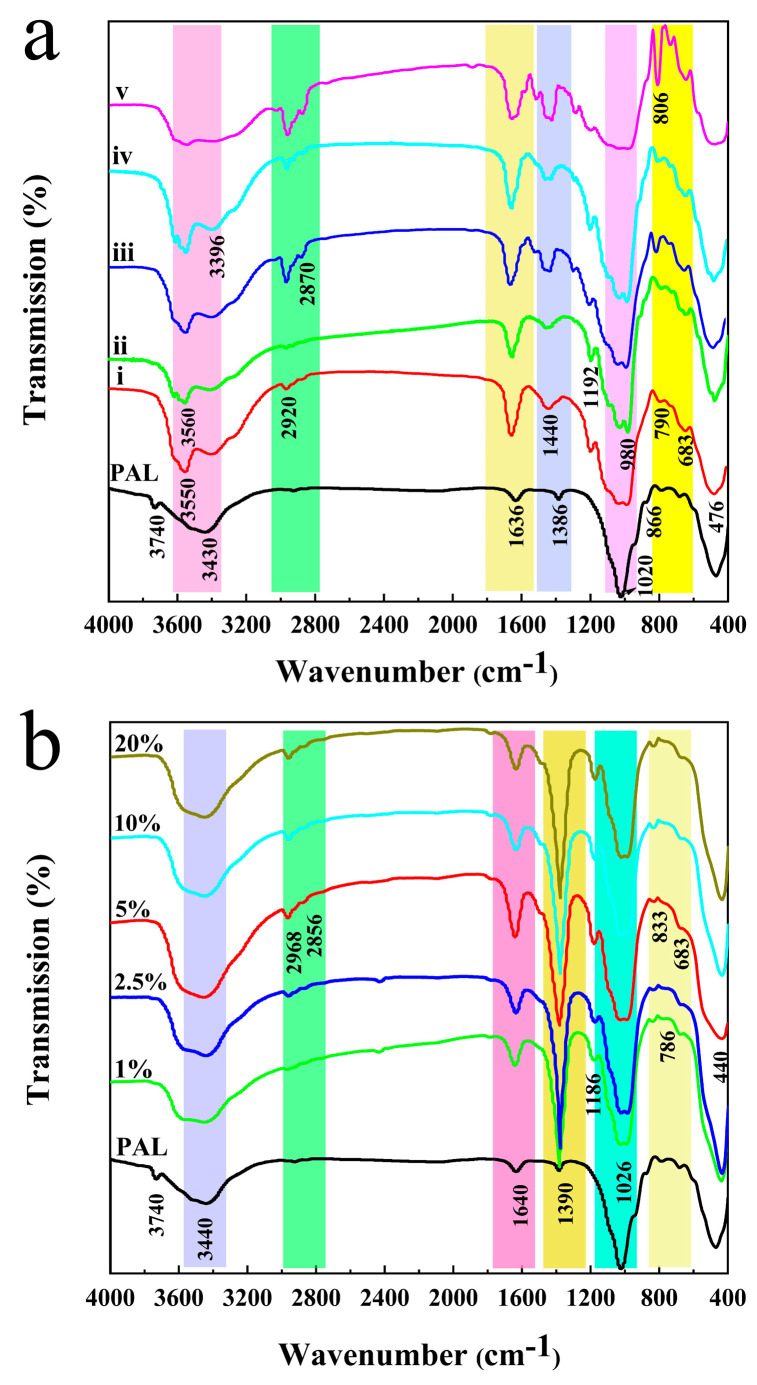
FTIR spectra of PEOs/ZnO/PAL (i) citral, (ii) thymol, (iii) CAR, (iv) oregano oil, (v) cinnamaldehyde, (**a**) and CAR/ZnO/PAL with different concentrations of CAR (**b**).

**Figure 3 nanomaterials-11-03230-f003:**
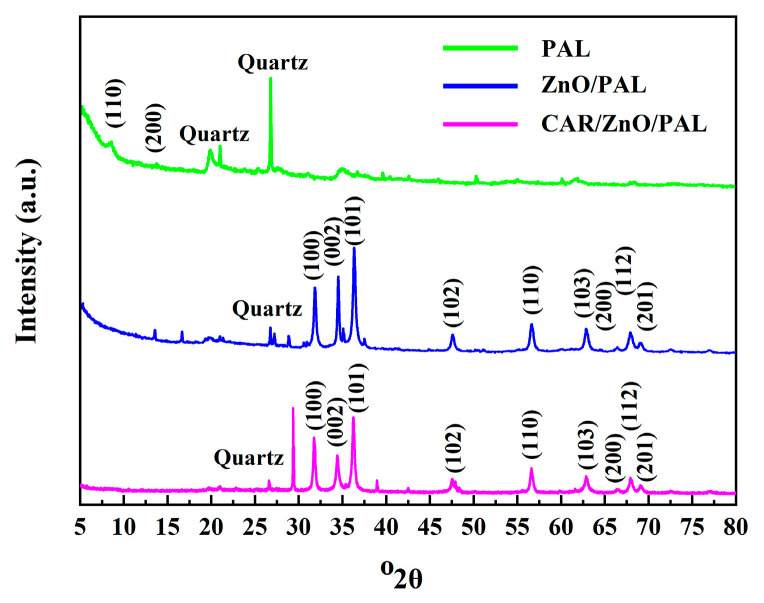
XRD pattern of PAL, ZnO/PAL, and CAR/ZnO/PAL.

**Figure 4 nanomaterials-11-03230-f004:**
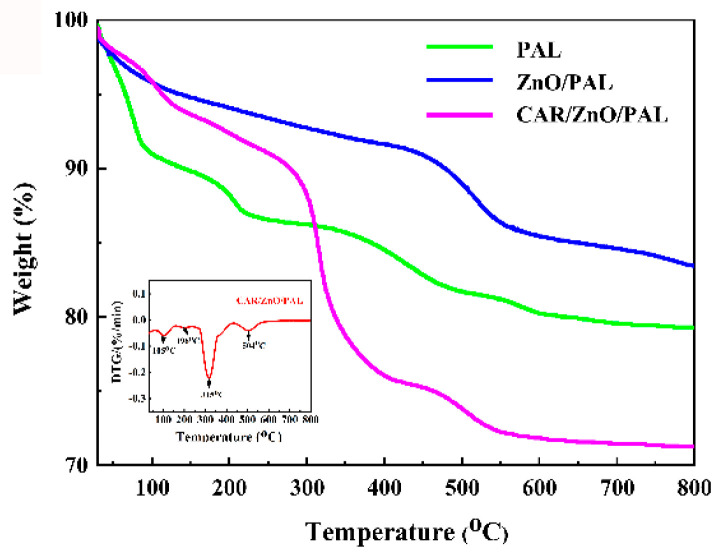
TG curves of PAL, ZnO/PAL, and CAR/ZnO/PAL.

**Figure 5 nanomaterials-11-03230-f005:**
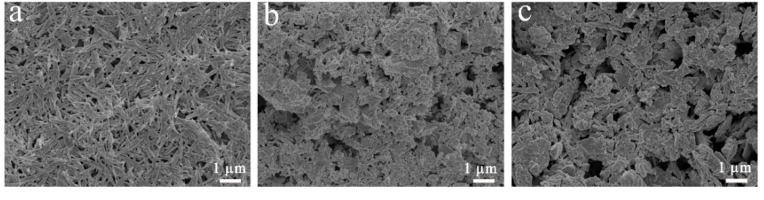
SEM images of (**a**) PAL, (**b**) ZnO/PAL, and (**c**) CAR/ZnO/PAL.

**Figure 6 nanomaterials-11-03230-f006:**
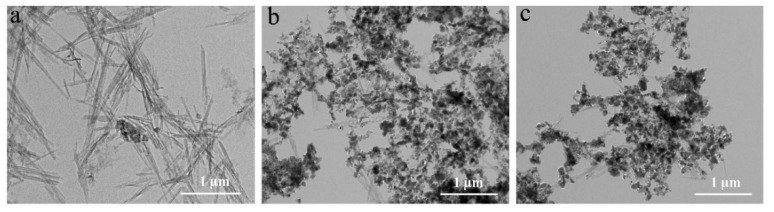
TEM images of (**a**) PAL, (**b**) ZnO/PAL, and (**c**) CAR/ZnO/PAL.

**Figure 7 nanomaterials-11-03230-f007:**
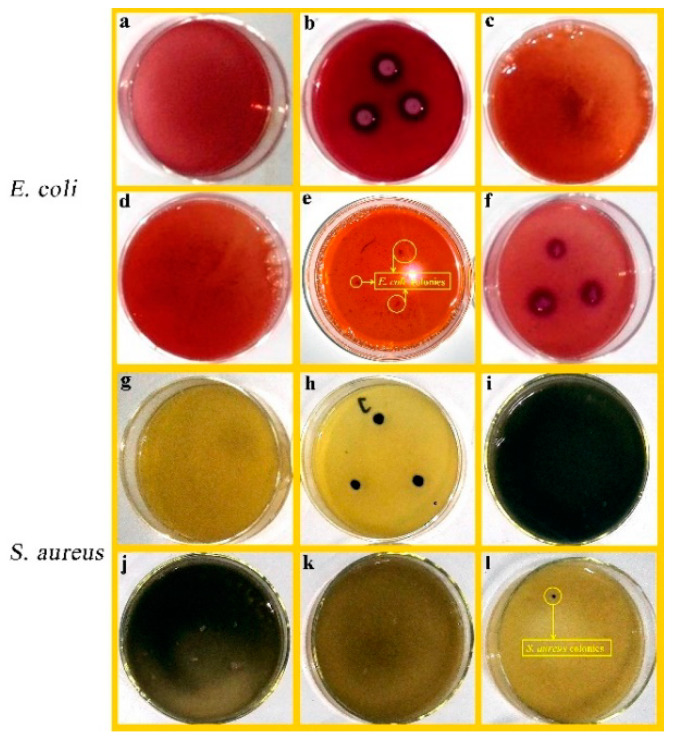
(**a**) Blank control, (**b**) positive control of *E. coli*, *E. coli* treated with ZnO/PAL nanoparticles at various concentrations of (**c**) 2.5 mg/mL, (**d**) 1.5 mg/mL, (**e**) 1 mg/mL, and (**f**) 0.5 mg/mL, (**g**) blank control, (**h**) positive control of *S. aureus*, *S. aureus* treated with ZnO/PAL nanoparticles at various concentrations of (**i**) 10 mg/mL, (**j**) 5 mg/mL, (**k**) 2.5 mg/mL, and (**l**) 1.5 mg/mL.

**Figure 8 nanomaterials-11-03230-f008:**
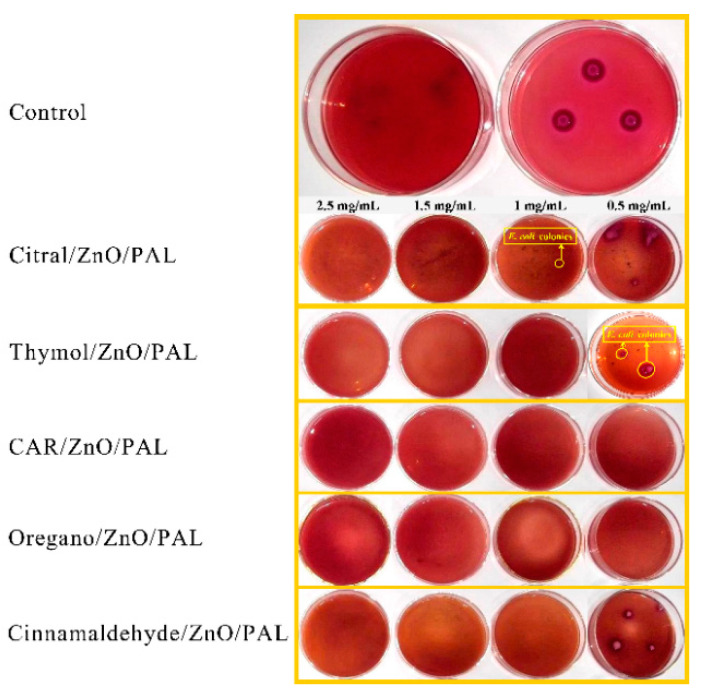
MIC value of PEOs/ZnO/PAL nanocomposites against *E. coli*.

**Figure 9 nanomaterials-11-03230-f009:**
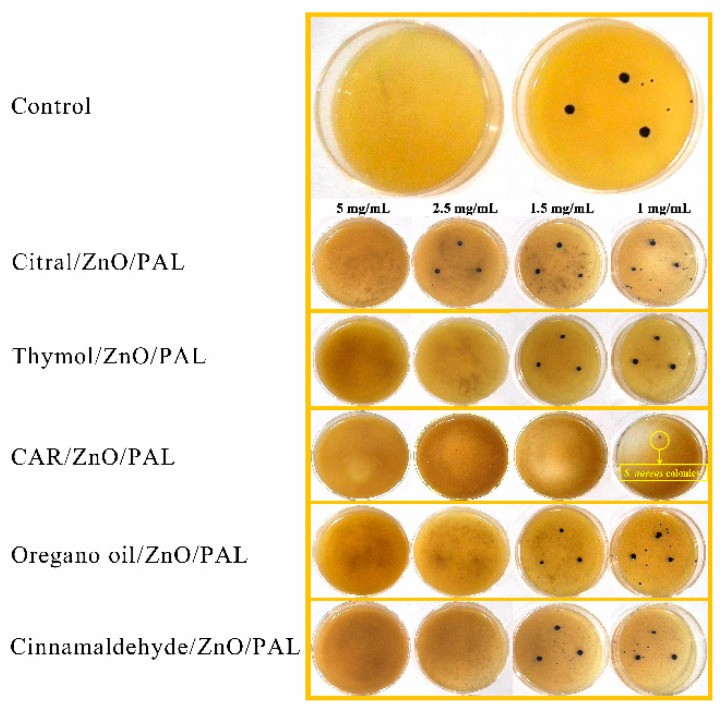
MIC value of PEOs/ZnO/PAL nanocomposites against *S. aureus*.

**Figure 10 nanomaterials-11-03230-f010:**
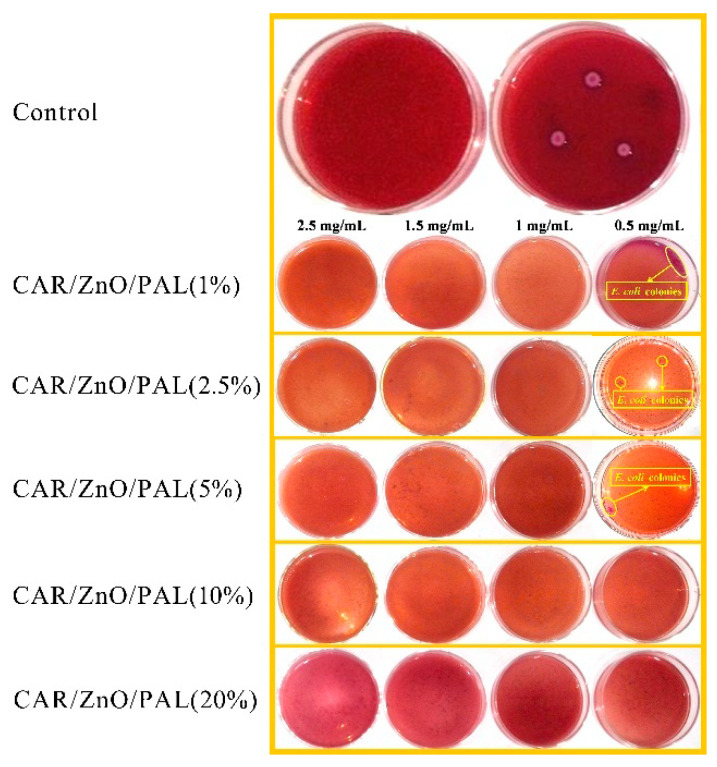
MIC evaluation of ZnO/PAL at different concentrations of CAR against *E. coli*.

**Table 1 nanomaterials-11-03230-t001:** *S*_BET_, *S*_micro_, *S*_ext_, and *V*_total_ of PAL, ZnO/PAL, and ZnO/PAL at different concentrations of CAR.

Concentrations of CAR (%)	*S*_BET_ (m^2^/g)	*S*_micro_ (m^2^/g)	*S*_ext_ (m^2^/g)	*V*_total_ (cm^3^/g)
1	25	−	29	0.149
2.5	19	−	21	0.133
5	15	−	17	0.091
10	13	−	16	0.082
20	9	−	11	0.034
PAL	125	8	117	0.332
ZnO/PAL	27	−	33	0.158

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
