# Peer review of "Palygorskite-Based Organic–Inorganic Hybrid Nanocomposite for Enhanced Antibacterial Activities"

_nanomaterials, 2021, doi:10.3390/nano11123230_

Round 1
Reviewer 1 Report
Dear Authors,
Although there are several important remarks to be considered, I find this work interesting.
The main remark is for thermogravimetry results. As first, these results should be separated under subtitle, not as a part of XRD results. Then, introducing also DTG, if possible, would make these results more informative. Next is water loss assignation. In fact it is not clear what the authors mean under „zeolite water “, „first half structural water“ and „second half structural water“, because this explanations in lines 150 to 156 does not seem correct.
Furthermore, BET results should be separated as another subtitle as well, not written as a part of SEM and TEM results.
In supplemental, in figs S2 and S3 there is no visible difference in the results obtained with different amount of PAL, therefore, please explain it.
Another important observation is that this manuscript needs some corrections and explanations to make this text easier to follow, and this should be corrected to ensure to the readers undisturbed reading and understanding of what is written. The remarks arewritten in the text following, consecutively from the beginning to the end of the manuscript:
-line 11, “however” with capital letter
-line 12 instead “characters” there should be “characteristics”
-line 14 after (ZnO/PAL) please write “nanoparticles“ or something like this to be clear
-line 16, „are“ is missing after ZnO nanoparticles
-line 22 sentence starting with „Therefore“ is not clear, please rewrite
-line 34 part of sentence starting with „and eventually cause“ does not match with the beginning of the sentence, it should be rewritten
-line 39, instead of comma it should be a new sentence in continuation
-lines 53 and 54 „higher aspect“ and „better“ in comparison to what?
-line 57 „Previous“ instead of „precious“
-lines 64 and 65 the sentence starting in line 64 is not clear, it should be rewritten
-line 77, after „use“ should start new sentence instead of comma
-line 95 “fluorescence”
-line 105 „a“ instead of „an“
-line 123 remove „as“ from the sentence
-line 140 In subtitle 3.2. is written XRD patterns, although there are also TG results. The TG results should be written as separate subtitle
-line 141 „are“ instead of „were“
-line 169 after“...PAL“ should start a new sentence instead of comma
-line 174 The sentence starting in this line is not clear and should be rewritten. Also „small molecules“ sounds better than „little molecules“
-line 182 here should be a new subtitle: BET
-line 183 The whole sentence starting in line 183 should be rewritten and divided in two or three sentences, it is not clear and correct as it is written now.
-line 184 remove word “was”
-line 186 insert “amount” after “increasing” to make this sentence clear
-line 194 molecules
-line 198, the sentence starting in this line is not clear, should be rewritten
-line 200 “by” should start with capital letter
-line 202 APT? Is this a typo?
-line 205 and 206 “broadening broad spectrum” does not sound well, it should be rewritten
-line 211 “were” instead of “was”
-line 214 this sentence is not clear, please rewrite it. Also it should be “was shown”
-line 217, “better” than what? The same also in lines 223 and 224
-line 239 The sentence starting in this line is not clear, please, rewrite it.
-line 245 it should be “as follows”
-line 259 the sentence starting in this line is not clear, it should be rewritten.
Regards
Author Response
1) Although there are several important remarks to be considered, I find this work interesting. The main remark is for thermogravimetry results. As first, these results should be separated under subtitle, not as a part of XRD results. Then, introducing also DTG, if possible, would make these results more informative. Next is water loss assignation. In fact it is not clear what the authors mean under zeolite water ''first half structural water" and, "second half structural water", because this explanations in lines 150 to 156 does not seem correct.
Response: Thanks for your suggestion. We have separated the TG and XRD results, and also added the DTG to explain the water loss assignation, and added the part 3.4. TG curves. Meanwhile, we have carefully checked these results and corrected the explanations in lines 150 to 156.
2) Furthermore, BET results should be separated as another subtitle as well, not written as a part of SEM and TEM results.
Response: Thanks for your comment. We have separated as another subtitle of BET results, and added a new part 3.6. BET analysis.
3) In supplemental, in figs S2 and S3 there is no visible difference in the results obtained with different amount of PAL, therefore, please explain it.
Response: Thanks for your suggestion. Figure S2 and Figure S3 are the results of PAL against gram-negative and gram-positive bacteria, these data as a control, to explain the PAL carrier was not antibacterial activities toward model bacteria, and the MIC (minimum inhibitory concentration) were more than 50 mg/mL. Meanwhile, there is no visible difference in the MIC results with different amount of PAL, PAL is not antibacterial effect toward Gram-negative E. coli and gram-positive S. aureus, that is, no particular selectivity or sensitivity. Some details have corrected in the manuscript.
4) Another important observation is that this manuscript needs some corrections and explanations to make this text easier to follow, and this should be corrected to ensure to the readers undisturbed reading and understanding of what is written. The remarks are written in the text following, consecutively from the beginning to the end of the manuscript.
Response: Thanks for your comment. We have corrected and added explanations to make this text easier to follow, the detailed modification one by one from the beginning to the end of the manuscript, and marked in red color.
Reviewer 2 Report
Currently, there are microorganism strains resistant to antibiotics and antiseptics. Therefore, researchers devote special attention to the search of new efficient antibacterial agents for biomedical applications. I recomment to accept this Manuscript after major revision.
- Figure 2a: Add phase designations tio figure.
- Figure 3: The quality of SEM image scale bars needs to improve.
- The quality of Figures 6-9 needs to improve. To determine the antibacterial activity (MIC value), a standard microdilution method could be used.
- It is essential to compare the antibacterial activity yur samples with other antibacterial agents.
Author Response
Currently, there are microorganism strains resistant to antibiotics and antiseptics. Therefore, researchers devote special attention to the search of new efficient antibacterial agents for biomedical applications. I recommend to accept this Manuscript after major revision.
1) Figure 2a: Add phase designations tio figure.
Response: Thanks for your comment. We have added phase designations tio figure of Figure 2.
2) Figure 3: The quality of SEM image scale bars needs to improve.
Response: Thanks for your comment. We have added new SEM image scale bars.
3) The quality of Figures 6-9 needs to improve. To determine the antibacterial activity (MIC value), a standard microdilution method could be used.
Response: Thanks for your comment. We have reprocessed the resolution and contrast to improve the definition of Figures 6-9. A standard microdilution method in this manuscript that we have republished in the previous work, we have cited this method, and listed in reference [31].
4) It is essential to compare the antibacterial activity yur samples with other antibacterial agents.
Response: Thanks for your suggestion. We have added the comparison of MIC value of palygorskite-based antibacterial materials as follows: Especially, compared with the previous antibacterial materials prepared by PAL carrier, the antibacterial activities of CAR/ZnO/PAL nanocomposites was superior to single ZnO/PAL nanoparticles (1.5 mg/mL) and CAR/PAL hybrid materials (2 mg/mL) toward gram-negative E. coli [31,43].
Reviewer 3 Report
This present work presented an interesting organic-inorganic hybrid nanocomposite materials showing high antibacterial activity towards E. coli and S. aureus. The authors have demonstrated the synthesis processes of nanocomposites and effort to uniform incorporation of ZnO nanoparticles. However, it is not enough to demonstrate the antibacterial effect of organic-inorganic hybrid materials to bacterial cells. Additionally, the manuscript should be revised and reorganize the figures. Therefore, the manuscript should be revised and resubmitted to this journal There is no Si-O-Si materials in this journal but the authors presented that FTIR is assigned to Si–O–Si band from the analysis. I think the authors should revised the manuscript and re-organize the manuscript. The Figure 5 should be figure 1, Figure 4 should be figure 2 then Figure 1 should be figure 3 for better understanding the meaning Is there any specific reason to choose CAR and analyze through the manuscript. There is no explain about Figure 5. The author indicated that five oil component but only present CAR composite materials. Is there any reason that why the author did not show any experimental results? I can not read the letters in Figure 6 and 7 and the images should be increased.Author Response
This present work presented an interesting organic-inorganic hybrid nanocomposite materials showing high antibacterial activity towards E. coli and S. aureus. The authors have demonstrated the synthesis processes of nanocomposites and effort to uniform incorporation of ZnO nanoparticles. However, it is not enough to demonstrate the antibacterial effect of organic-inorganic hybrid materials to bacterial cells. Additionally, the manuscript should be revised and reorganize the figures. Therefore, the manuscript should be revised and resubmitted to this journal. There is no Si-O-Si materials in this journal but the authors presented that FTIR is assigned to Si–O–Si band from the analysis. I think the authors should revised the manuscript and re-organize the manuscript. The Figure 5 should be figure 1, Figure 4 should be figure 2 then Figure 1 should be figure 3 for better understanding the meaning Is there any specific reason to choose CAR and analyze through the manuscript. There is no explain about Figure 5. The author indicated that five oil component but only present CAR composite materials. Is there any reason that why the author did not show any experimental results? I can not read the letters in Figure 6 and 7 and the images should be increased.
Response: Thanks for your suggestion. We have revised and re-organized the manuscript according to the Reviewers comments. The figure of manuscript have revised for better understanding the meaning. FTIR is assigned to Si–O–Si band from the analysis, the main reason is palygorskite structure after being calcined, these results referenced from cited paper of reference [40]. We have added some reason to choose CAR and analyze through the manuscript, and also have explained the Figure 5. In the manuscript, Figure 6 is a comparative experimental results to evaluate the MIC value of ZnO/PAL nanoparticles. Figure 7 and 8 is the MIC results of PEOs/ZnO/PAL nanocomposites against E. coli and S. aureus, to investigate the relationship between ZnO/PAL nanoparticles and five PEOs, to obtain the superior combination effect and antibacterial activities.
In summary, the English grammar and expression have been carefully checked and corrected during revision according to the reviewer’s suggestion. Furthermore, we have re-organized the manuscript according to the three Reviewers comments.
Round 2
Reviewer 2 Report
I recommend to accept this articles in present form
Reviewer 3 Report
The manuscript has been improved and now it is ready to publish in this journal